# Novel Histologic Categorization Based on Lauren Histotypes Conveys Prognostic Information for Gastroesophageal Junction Cancers—Analysis from a Large Single Center Cohort in Germany

**DOI:** 10.3390/cancers13061303

**Published:** 2021-03-15

**Authors:** Rebekka Schirren, Alexander Novotny, Julia Slotta-Huspenina, Helmut Friess, Daniel Reim

**Affiliations:** 1Department of Surgery, TUM School of Medicine, Ismaninger Strasse 22, 81675 Munich, Germany; rebekka.schirren@tum.de (R.S.); alexander.novotny@tum.de (A.N.); helmut.friess@tum.de (H.F.); 2Institute of Pathology, TUM School of Medicine, Ismaninger Strasse 22, 81675 Munich, Germany; julia.slotta-huspenina@tum.de

**Keywords:** gastric/gastroesophageal cancer, perioperative chemotherapy, Lauren histotype

## Abstract

**Simple Summary:**

The incidence of adenocarcinomas of the GE-junction (AEG) increases evermore over the recent decades in the Western world. The postoperative tumor stage with or without multimodal treatment provides important information with regard to post-therapeutic survival. However, this tumor staging may be influenced by the histologic tumor subtype. These subtypes were described by Lauren before and provide substantial prognostic information in gastric cancer patients. The Lauren classification, however, was not yet evaluated in AEG before the purpose of the present retrospective analysis. It was confirmed in a cohort of 1153 AEG patients that Lauren subtypes convey substantial prognostic information after surgical resection. Furthermore, a simplified sub-classification into differentiated and undifferentiated histotypes was developed and evaluated. This newly proposed sub-classification system requires further confirmation by multicentric re-evaluation analyses.

**Abstract:**

Adenocarcinoma of the gastroesophageal junction (AEG) ranks among the most common cancers in the Western world with increasing incidence. However, the prognostic influence and applicability of the Lauren classification was not examined in detail before. The purpose of this analysis was to analyze the oncologic outcomes of GE-junction cancer related to the Lauren histotype in a large single center cohort. Data from the prospectively documented database of the Klinikum Rechts der Isar (TUM School of Medicine) for patients undergoing curatively intended oncologic resection for GE-junction cancer between 1984 and 2018 were extracted. Univariate and multivariate regression analyses were performed to identify predictors for overall survival. Kaplan-Meier analyses were done to investigate the survival rates according to the Lauren histotype. After identification of two distinct histologic categories with prognostic implications, propensity score matching (PSM) was performed to balance for confounders and evaluate its oncologic outcomes retrospectively. In the time period indicated, 1710 patients were treated for GE-junction cancer. Exclusion criteria were: R2-resections (*n* = 134), metastatic disease (*n* = 296), 30-day mortality (*n* = 45), Siewert type I (*n* = 21), and missing/incomplete data (*n* = 61). Finally, 1153 patients were analyzed. In a multiple variable analysis, age, UICC-stage, all Lauren histotypes, R-stage, and postoperative complications were significant predictors of overall survival. Kaplan Meier analysis demonstrated significant survival differences between intestinal, diffuse, and mixed Lauren-histotypes (*p* = 0.001 and *p* = 0.029). Survival rates were comparable between non-classifiable and intestinal Lauren-types (*p* = 0.16) and between diffuse and mixed types (*p* = 0.56). When combining non-classifiable, well, and moderately differentiated Lauren-types and combining poorly differentiated intestinal, diffuse, and mixed types, two highly prognostic groups were identified (*p* < 0.0001). This was confirmed after PSM for possible confounders. The Lauren histotypes demonstrate highly prognostic value after oncologic resection of GE-junction cancer (Siewert type II and type III) in a single center Western patient cohort. A simplified histotype classification based on Lauren subtypes revealed a clear distinction of prognostic groups and should be considered for further evaluation.

## 1. Introduction

Adenocarcinomas of gastroesophageal junction (AEG) are an increasing tumor entity in the western world [1] in contrast to Eastern Asia. The Lauren classification is the most common histological classification of gastric cancer and distinguishes between types: intestinal, diffuse, and mixed types [2]. Another commonly applied histologic classification was published by the World Health Organization (WHO) [3]. Part of the current scientific discussion is a unification of the different classifications [4]. With regard to gastric cancer, it has been shown in recent years that the Lauren classification has prognostic impact. Furthermore, it was shown that Lauren subtypes respond differently to chemotherapy [5,6,7,8,9,10]. However, this was demonstrated in mostly retrospective patient data, since large therapy studies were not adequately powered for subgroup analyses regarding the different Lauren subtypes [11,12,13,14]. Since the tumors of the gastroesophageal junction (GEJ) are partly considered gastric carcinomas, the Lauren classification is traditionally applied here as well. Several analyses were carried out in recent years to determine whether tumors of the GEJ should be classified as gastric carcinomas or esophageal carcinomas. In this context, both molecular biological patterns and prognostic behaviors were researched before [15,16,17]. Hitherto, no clear consensus was reached if AEG belongs to gastric cancer, esophageal cancer, or if they even represent an own entity. However, to the authors’ knowledge, no validation to apply the Lauren classification in AEG for prognostic implications was carried out yet. Since neither the applicability nor the prognostic value of the Lauren classification has been investigated in AEG yet, the authors aimed to clarify these issues by analyzing a large single center cohort from a specialized tertiary treatment center.

## 2. Results

1710 patients who were treated for GE-junction malignancy were identified from the institutional database in a period from 1984 to 2018. Finally, after removing patients meeting the exclusion criteria (R2-resection (*n* = 134), metastatic disease (*n* = 296), 30-day mortality (*n* = 45), Siewert type I (*n* = 21), and missing/incomplete data (*n* = 61)) 1153 patients were available for analysis. These were 666 patients with a Lauren intestinal type, 172 patients of a Lauren diffuse type, 127 patients of a Lauren mixed type, and 188 patients who could not be classified according to Lauren. The analysis of the baseline characteristics showed significant differences between Lauren subtypes regarding gender distribution, comorbidities, tumor-localization (diffuse type was more frequent in Siewert type III), neoadjuvant treatment, and surgery type. There were more advanced pT-stages and UICC-stages in Lauren diffuse types and mixed types. Well and moderately differentiated types were more frequent for intestinal and non-classified types, and higher R1 rates were identified for diffuse and mixed subtypes. No significant differences were found regarding D2-lymphadenectomy rates, age distributions, and postoperative complication rates (Table 1).

Median follow-up was 34 months (range 1–273 months), comprising of 79 months (range 1–273 months) for survivors and 22 months (range 1–219) for deceased patients. During the follow-up period, 664 patients (57.6%) died, the five-year survival rate (FYSR) was 44%, and the ten-year survival rate (TYSR) was 26%. Median survival was 90 months for patients with a non-classifiable Lauren type (55/37% FYSR/TYSR), 58 months for intestinal type (45/27% FYSR/TYSR), 31 months for diffuse type (34/17% FYSR/TYSR), and 38 months for a mixed Lauren type (37/24% FYSR/TYSR).

Kaplan Meier analysis revealed no statistically significant survival differences between non-classifiable and intestinal G1 and G2 subtypes (HR 1.0 vs. HR 1.02, CI95% 0.69–1.51, *p* = 0.93 for non-classifiable vs. intestinal G1 type, HR 1.02, CI95% 0.69–1.51 vs. HR 1.04, CI95% 0.80–1.35, *p* = 0.92 for intestinal G1 type vs. intestinal G2 types). Patients with poorly differentiated intestinal types (G3), diffuse types, and mixed types did not show significant survival differences between each other (HR 1.43, CI95% 1.10–1.86 vs. HR 1.68, CI95% 1.27–2.23, *p* = 0.18, G3 intestinal vs. diffuse and HR 1.68, CI95% 1.27–2.23 vs- HR 1.55, CI95% 1.14–2.11, *p* = 0.56, diffuse vs. mixed types, Figure 1). Based on these observations, these subgroups were summarized into two prognostically relevant groups (simplified Lauren types): Differentiated type (non-classifiable, intestinal G1 and intestinal G2 types) vs. undifferentiated type (intestinal G3, diffuse, and mixed type). These groups were statistically significant prognosticators of survival (HR 1.49, CI95% 1.28–1.73, *p* < 0.0001, Figure 2).

In order to analyze independent survival predictors, the following variables were included in the cox regression analysis: Age, Siewert type, gender, neoadjuvant therapy, Lauren subtype, lymph node dissection, UICC stage, R-status, grading, comorbidity, and complications. pT-stages and pN-stages were not included since these factors were cumulated in the UICC stage. All factors were entered in the multivariate model without selections. Univariate regression analysis revealed age, localization, Lauren histotype, D2-dissection, UICC stage, R-status, grading, and occurrence of post-operative complications to be significantly associated with post-therapeutic survival. The multiple variable analysis showed that age, Lauren subtype, UICC stage, R-status, and postoperative complications were independent survival predictors (Table 2).

After identification of two distinct subtypes (differentiated/undifferentiated), baseline characteristics were re-analyzed. In the comparison of both groups, there were significant differences for the following baseline characteristics: Gender, comorbidities, tumor localization, type of surgery performed, pT-/pN-/UICC-stages, grading, and R0 resection rates. Age, type, and number of patients receiving neoadjuvant therapies, the number of dissected lymph nodes, and postoperative complication rates were equally distributed (Table 3). In order to overcome potential confounders due to these differences, the two cohorts were matched by adjusting for these potentially confounding factors: Gender, localization, and R-status.

### Results after Propensity Score Matching (PSM)

Those variables demonstrating clinically meaningful baseline differences within the respective Lauren subgroups were matched through PSM (gender, location, R-status) to balance possible confounders. The matching algorithm matched 471 patients each in the differentiated and undifferentiated subtype groups. Analysis of the baseline characteristics demonstrated that the following variables were then well balanced in all groups: gender, age distribution, comorbidities, tumor localization, neoadjuvant therapies, D2 dissection rate, post-operative complications, and R0 status. The results are shown in Table 3 and Table 4. The post PSM balancing analysis revealed adequate matching of the respective variables (covariate balancing plots and matching data are shown in Appendix A).

Median follow-up was 35 months (range 1–273 months), comprising of 83 months (range 1–273 months) for survivors and 22 months (range 1–218) months for deceased patients (*p* < 0.0001). During the follow-up period, 560 patients (58.5%) died, the FYSR was 44%, and the TYSR was 26%. Median survival was 67 months for patients with a differentiated histotype and 37 months for patients with an undifferentiated subtype (HR 1.31, CI95% 1.11–1.54, *p* = 0.002) (Figure 2). FYSR/TYSR for differentiated/undifferentiated subtypes were 49/29% and 38/23%, respectively.

The UICC stage-dependent analysis revealed no significant survival differences for UICC stages I and II (UICC I: median survival not met for differentiated type vs. 216 months in undifferentiated types, HR 0.79, CI95% 0.5–1.27, *p* = 0.33, Figure 3, UICC II: median survival 60 months (differentiated type) vs. 67 months (undifferentiated type), HR 1.00, CI95% 0.73–1.36, *p* = 0.996, Figure 4). In UICC III, there was a statistically significant survival difference between differentiated and undifferentiated types. Median survival was 25 months (differentiated type) vs. 16 months (undifferentiated type) (HR 1.32, CI95% 1.05–1.64, *p* = 0.016) (Figure 5). FYSR and TYSR were 26/7% in differentiated types vs. 15/10% in undifferentiated types in the UICC III stage.

The univariate analysis of the PSM-cohort demonstrated that age, undifferentiated type, UICC-stage, R1-resection, grading, and occurrence of postoperative complications were significantly related to overall survival. In the multivariate model age, undifferentiated type and UICC-stage were demonstrated to be independent prognostic variables with regard to overall survival (Appendix A). 

## 3. Discussion

This retrospective analysis of a large single center patient cohort of 1153 patients demonstrates that the Lauren subtype is a significant and independent prognostic variable related to overall survival of adenocarcinoma at the gastroesophageal junction. Besides this, a novel simplified categorization of the Lauren subtypes into a differentiated and undifferentiated category was demonstrated to be independently and significantly related to overall survival. This was demonstrated by multivariate cox regression and Kaplan Meier analyses. The analysis of the respective Lauren subtypes showed that a simplified model can be formed due to similar plotting of the respective survival curves. It was shown that intestinal subtypes with good and moderate differentiation and non-classifiable tumors were comparable regarding survival outcome and were, therefore, pooled in a group of a differentiated category. Poorly differentiated intestinal types, diffuse types, and mixed types demonstrated (statistically) similar survival outcomes and were, therefore, summarized for the undifferentiated category, which proved to be prognostically significant in a propensity score matched cohort.

It was not yet shown that the Lauren classification would be applicable for AEG. This analysis revealed that the Lauren subtypes convey important prognostic implications similar to gastric cancer patients for AEG, which is predominantly evident in the Western world. The Lauren classification is a histological classification introduced in 1965 for adenocarcinoma of the stomach and defines three subtypes: intestinal, diffuse, and mixed type adenocarcinomas [2]. Another simplified version was proposed by Nakamura in 1968, which resembles the two histologic categories in the present analysis [18]. Nonetheless, the Nakamura classification was not yet evaluated for applicability in AEG either. The important finding of the present analysis is that there are marked differences between the novel proposed categories. The predominant proportion of male patients is reduced in patients with the undifferentiated category. The undifferentiated types were more located in the sub-cardial parts of the GE-junction and differentiated types revealed to be less advanced (in terms of UICC, pT-stages, and pN-stages) in the histologic workup after resection, and R0 resection rates were higher in the differentiated category. The differences regarding UICC-stages were confirmed after the possibly confounding factors were balanced by PSM, creating comparable groups with regard to gender, localization, and R-staging. The rationale for applying PSM was based on literature recommendations. Most experts in the field of PS-matching describe four relevant methods to estimate treatment effects: matching, stratification (or subclassification), weighting, and covariate adjustment using the propensity score. Weighting and covariate adjustment were ruled out in the setting of the present analysis because these methods use the propensity score directly for estimating the effect of treatment. PSM and stratification use the propensity score for grouping subjects but not estimating the effect of treatment, which was intended here (grouping patients without estimating treatment effects). Furthermore, it was described that weighting and covariate adjustments may be more sensitive to misspecification of the propensity score model than matching and stratification [19,20,21,22]. It was not intended to balance for UICC-stages in order to describe the biologic behaviors of the novel proposed and simplified histologic categories. Therefore, it was shown that, in the stage-dependent survival analysis, the prognostic effect of the simplified categories was evident only in UICC III. Taking these observations into consideration, it can be stated that, in earlier UICC-stages, the Lauren classification or simplified histotypes do not have the same prognostic relevance as in more advanced stages. The present data further imply that the cancer progression is biologically more aggressive in the undifferentiated category. The relevance for clinical practice would be to provide narrower follow-up intervals to these patients in order to detect early recurrence or cancer progression. It remains elusive if different or more aggressive chemotherapy protocols in the perioperative setting should be offered to those patients in the undifferentiated histologic category. It may be speculated that most of the chemotherapy regimens presently available may be effective at all, as it was shown before that poorly cohesive tumors/tumors with signet ring-cell-like histology in gastric cancer do not respond very well to currently approved chemotherapy regimens, such as FLOT [5,8,23,24]. This is also reflected by the results of the multiple variable analysis in the present cohort, which failed to demonstrate a beneficial effect.

Certainly, this analysis has multiple limitations, which are not only the monocentric and retrospective character, but also the long observation period during which both surgical and perioperative regimens have changed. Although potential biases inherent to the different baseline characteristics of the two histologic categories were possibly corrected by PSM, this method cannot compensate for unconscious and biological biases or for undetermined factors. More than that, it is critical that the PSM resulted in a smaller number of patients per group than in the primary analysis. Therefore, no exact statements can be made about unmatched patients. Another limitation is that the PS-matching did not balance adequately for the UICC-stages, so that the balance is skewed toward more advanced cases in the undifferentiated subtype group, which might limit further conclusions regarding survival prognosis. Furthermore, generalizability of the present results is certainly restricted, as AEG is predominantly evident in the Western world and the findings are not transferable to Asian patients due to ethnicity and more importantly due to the fact that AEG incidence is still considerably low in countries such as Korea, Japan, and China. Besides this, perioperative chemotherapeutic regimens have changed over time and, therefore, its effect was not studied in this analysis.

Taking these limitations and findings into consideration, a future study should focus on multicentric evaluations of the proposed simplified histologic categories. In future prospective studies, this model should be further evaluated to better assess its significance with regard to therapeutic decisions.

## 4. Materials and Methods

### 4.1. Patients

The prospectively documented gastric cancer database at the surgical Department of TUM, Munich, Germany was screened for patients with adenocarcinoma of the GE-junction (AEG) undergoing surgery between 1984 and 2018 to identify eligible patients for this retrospective analysis. Clinical data was entered into the database after discharge and based on patient charts and records. Inclusion criteria were: Histologically proven GE-junction cancer (Siewert type II and III), curatively intended resections (R0/R1), and a documented histological type, according to Lauren. Exclusion criteria were: R2 resections (*n* = 134), metastatic disease (*n* = 296), 30-day mortality (*n* = 45), Siewert Type I (*n* = 21), and missing or incomplete data (*n* = 61). The patient inclusion flow chart is depicted in Appendix A.

All patients underwent multidisciplinary team review ahead of treatment after staging was performed by endoscopy, an endoscopic ultrasound, and a CT scan. For locally advanced cancer, multimodal therapy (neoadjuvant/perioperative chemotherapy or radiotherapy) was recommended based on the German S3-guideline. Neoadjuvant/perioperative treatment consisted of either two preoperative cycles of cisplatin or oxaliplatin/leucovorin/5-FU (PLF/OLF) or three pre-operative and post-operative cycles of ECX/ECF (MAGIC) or four pre-operative and post-operative cycles of FLOT. In case of chemoradiation, the CROSS protocol (Ref) was applied. All surgical procedures were performed according to the Japanese gastric cancer treatment guideline including D2-lymphadenectomy. The surgical procedure was extended to the distal esophagus if intra-operative frozen sections demonstrated an oral margin infiltration until an intra-operative R0 situation was confirmed by a frozen section. In case of tumor extension of more than 5 cm (as determined by clinical staging) into the distal esophagus, an Ivor-Lewis procedure was performed including a two-field lymphadenectomy. All resected specimens were examined by one or two specialized pathologists, classified according to the TNM-classification and staged according to UICC-recommendations (8th edition). Patients were followed for 60 months from the day of surgery every six to twelve months in a dedicated outpatient department (Roman Herzog Comprehensive Cancer Center) by endoscopy and CT scans, according to the institutional protocol. Survival data was collected based on either additional visits or phone contacts. The dataset consisted of patients’ gender, age, tumor location, application of neoadjuvant chemoradiation or chemotherapy, type of surgery (esophagectomy, gastrectomy with transhiatal extension, gastrectomy), type of required extension (none, luminal/transhiatal, splenectomy, colon, pancreas, others), number of dissected lymph nodes, D2 lymphadenectomy success rate, postoperative complications (none, Clavien–Dindo Grade I/II and III/IV), pT-(pT1/pT2/pT3/pT4), pN-(pN0/pN1/pN2/pN3a), and UICC-stages (UICC-I/-II/-III), grading (G1/2, G3/4), R-status (R0/R1), Lauren histotype (intestinal, diffuse, mixed, unclassifiable), and follow-up period with a survival status.

### 4.2. Statistical Analyses

Descriptive statistics on demographic and clinical tumor characteristics were calculated as the mean ± standard deviation (continuous variables) and frequencies (categorical variables). Survival time was calculated from the day of surgery to death or last follow-up date. The Kaplan-Meier method was used to estimate survival probabilities stratified by the application of neoadjuvant/perioperative chemotherapy. The log-rank test was used to compare the estimated survival. Survival prognosticators were analyzed by univariate and multivariate cox regression analyses. Variables entered into the model were age, Siewert type II and III, gender, neoadjuvant chemotherapy, Lauren subtype (differentiated/undifferentiated type in the PSM cohort), D2 dissection, UICC-stage, R-status, grading, comorbidity, and postoperative complications. After univariate analysis, all variables were entered in the multivariate model. After the primary analysis, the two groups (differentiated/undifferentiated type) demonstrated marked baseline differences, which were balanced for the clinically most relevant confounders (Gender, localization, R-status). The groups were matched by a “nearest neighbor” 1:1 matching with a 0.1 caliper. After PSM, data from 958 patients were reanalyzed after exclusion of 195 patients. All statistical analyses were performed using SPSS version 25 (IBM Inc., Ehningen, Germany). *p*-values less than 0.05 were considered statistically significant. This retrospective analysis was approved by the local Instituional Review Board (IRB) (No.364/20s; Ethikkommission der Fakultät für Medizin, TUM School of Medicine).

## 5. Conclusions

In conclusion, the present findings demonstrate that Lauren subtypes might be relevant prognostic factors in gastroesophageal cancer outcomes. Data from this analysis suggest that there might be two different histopathologic categories (differentiated/undifferentiated) with a prognostic impact. These proposed categories require further validation in multicenter cohorts. Their prognostic abilities to predict outcomes of neoadjuvant therapies are to be evaluated.

## Figures and Tables

**Figure 1 cancers-13-01303-f001:**
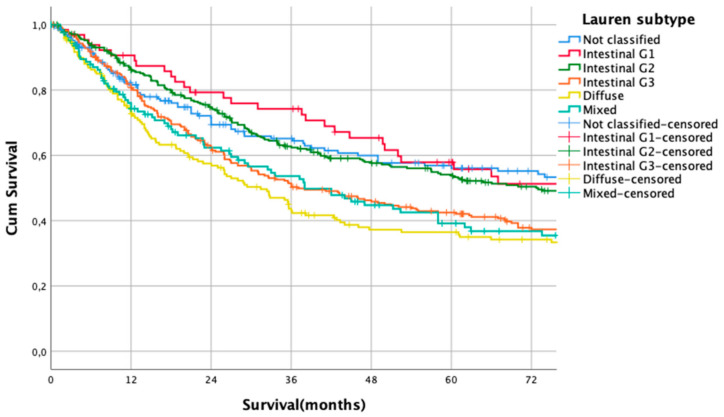
Survival curves for Lauren subtypes. HR 1.0 vs. HR 1.02, CI95% 0.69–1.51, *p* = 0.93 for non-classifiable vs. intestinal G1 type, HR 1.02, CI95% 0.69–1.51 vs. HR 1.04, CI95% 0.80–1.35, *p* = 0.92 for intestinal G1 type vs. intestinal G2 types. HR 1.43, CI95% 1.10–1.86 vs. HR 1.68, CI95% 1.27–2.23, *p* = 0.18. G3 intestinal vs. diffuse and HR 1.68, CI95% 1.27–2.23 vs. HR 1.55, CI95% 1.14–2.11, *p* = 0.56, diffuse vs. mixed types.

**Figure 2 cancers-13-01303-f002:**
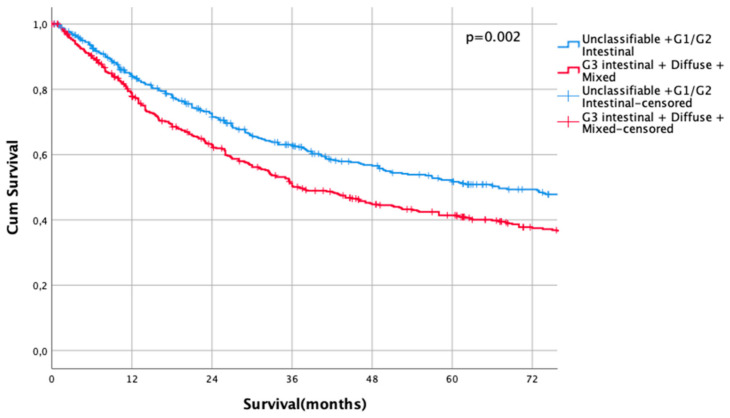
Survival curves for differentiated/undifferentiated subtypes after propensity score matching (PSM). Differentiated type (unclassifiable Lauren type, Lauren intestinal type G1 and G2) vs. Undifferentiated type (Lauren intestinal type, diffuse type, and mixed type). HR 1.31, CI95% 1.11–1.54, *p* = 0.002.

**Figure 3 cancers-13-01303-f003:**
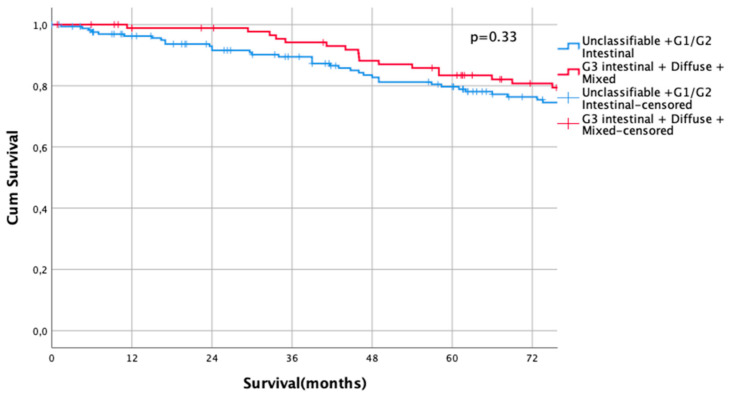
Survival curves for differentiated/undifferentiated subtypes after propensity score matching (PSM) in UICC I stage. Survival UICC I (*n* = 260) for a differentiated type (unclassifiable Lauren type, Lauren intestinal type, G1 and G2) vs. undifferentiated type (Lauren intestinal type, diffuse type, and mixed type). HR 0.79, CI95% 0.5–1.27, *p* = 0.33.

**Figure 4 cancers-13-01303-f004:**
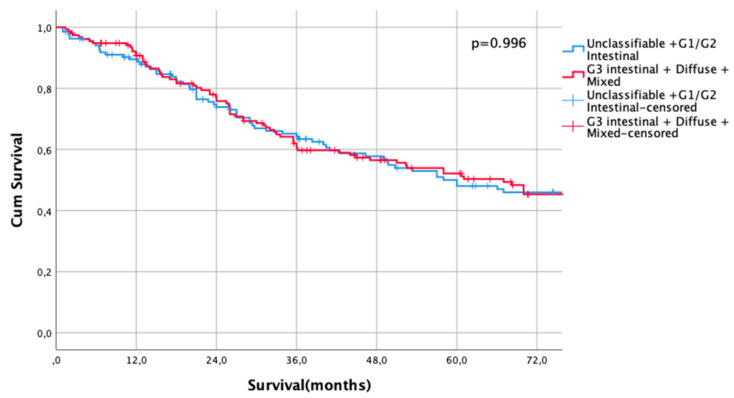
Survival curves for differentiated/undifferentiated subtypes after propensity score matching (PSM) in UICC II stage. Survival UICC II (*n* = 291) for a differentiated type (unclassifiable Lauren type, Lauren intestinal type G1 and G2) vs. undifferentiated type (Lauren intestinal type, diffuse type, and mixed type). HR 1.00, CI95% 0.73–1.36, *p* = 0.996.

**Figure 5 cancers-13-01303-f005:**
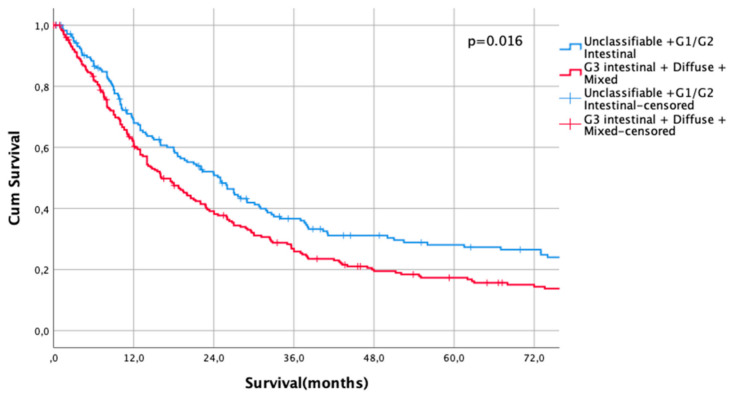
Survival curves for differentiated/undifferentiated subtypes after PSM in UICC III stage. Survival UICC III (*n* = 407) for a differentiated type (unclassifiable Lauren type, Lauren intestinal type G1 and G2) vs. undifferentiated type (Lauren intestinal type, diffuse type, and mixed type). HR 1.32, CI95% 1.05–1.64, *p* = 0.016.

**Table 1 cancers-13-01303-t001:** Baseline characteristics for patients with respective Lauren subtypes before propensity score matching (PSM).

Lauren Type	Unclass. (*n* = 188)		Int. (*n* = 666)		Diff. (*n* = 172)		Mixed (*n* = 127)		*p*-Value
	*n*	%	*n*	%	*n*	%	*n*	%	
Gender									**<0.0001**
Female	32	17.02	124	18.62	64	37.21	36	28.35	
Male	156	82.98	542	81.38	108	62.79	91	71.65	
Age									
<70	130	69.15	427	64.11	123	71.51	90	70.87	0.15
>70	58	30.85	239	35.89	49	28.49	37	29.13	
Comorbidities									0.003
None	155	82.45	455	68.32	119	69.19	88	69.29	
Single	10	5.32	111	16.67	30	17.44	23	18.11	
Multiple	23	12.23	100	15.02	23	13.37	16	12.60	
Localization									<0.0001
Siewert II	116	61.70	387	58.11	55	31.98	58	45.67	
Siewert III	72	38.30	279	41.89	117	68.02	69	54.33	
Neoadjuvant									<0.0001
Chemotherapy	109	57.98	267	40.09	71	41.28	50	39.37	
Chemoradiation	8	4.26	10	1.50	4	2.33	1	0.79	
None	71	37.77	389	58.41	97	56.40	76	59.84	
Type of Surgery									<0.0001
Esophagectomy	62	32.98	93	13.96	20	11.63	20	15.75	
Ext. Gastrectomy	103	54.79	481	72.22	125	72.67	96	75.59	
Total gastrectomy	9	4.79	52	7.81	27	15.70	10	7.87	
Merendino	14	7.45	40	6.01	0	0.00	1	0.79	
Surgical extension									<0.0001
None	67	35.64	158	23.72	44	25.58	36	28.35	
Luminal/transhiatal	81	43.09	274	41.14	67	38.95	48	37.80	
Splenectomy	11	5.85	41	6.16	17	9.88	13	10.24	
Colon	4	2.13	1	0.15	2	1.16	1	0.79	
Pancreas	1	0.53	21	3.15	1	0.58	2	1.57	
Others	24	12.77	171	25.68	41	23.84	27	21.26	
Diss. LN (Median)	27 (3–74)		29 (3–218)		31 (9–104)		32 (3–76)		0.09
<=25	72	38.30	232	34.83	46	26.74	39	30.71	
>25	116	61.70	434	65.17	126	73.26	88	69.29	
Complications									0.96
None	138	73.40	483	72.52	127	73.84	90	70.87	
CD I/II	30	15.96	95	14.26	23	13.37	19	14.96	
CD III-V	20	10.64	88	13.21	22	12.79	18	14.17	
pT									**<0.0001**
pT1	52	27.66	154	23.12	14	8.14	11	8.66	
pT2	16	8.51	99	14.86	11	6.40	18	14.17	
pT3	82	43.62	296	44.44	66	38.37	61	48.03	
pT4	38	20.21	117	17.57	81	47.09	37	29.13	
pN									**<0.0001**
pN0	84	44.68	294	44.14	69	40.12	37	29.13	
pN1	29	15.43	122	18.32	18	10.47	20	15.75	
pN2	35	18.62	111	16.67	26	15.12	18	14.17	
pN3	40	21.28	139	20.87	59	34.30	52	40.94	
UICC									**<0.0001**
UICC I	59	31.38	207	31.08	21	12.21	21	16.54	
UICC II	52	27.66	203	30.48	57	33.14	33	25.98	
UICC III	77	40.96	256	38.44	94	54.65	73	57.48	
Grading									**<0.0001**
G1/G2	33	17.55	335	50.30	2	1.16	7	5.51	
G3/G4/Gx	155	82.45	331	49.70	170	98.84	120	94.49	
R									**<0.0001**
R0	161	85.64	617	92.64	136	79.07	107	84.25	
R1	27	14.36	49	7.36	36	20.93	20	15.75	

Legend: pT1 = Mucosa/Submucosa. pT2 = Muscularis propria. pT3 = Serosa. pT4 = Adjacent organs. pN0 = no lymph nodemetastasis detected during staging. pN1 = 1–2 locoregional lymph node metastasis evident. pN2 = 3–7 locoregional lymph node metastasis evident. pN3 >15 locoregional lymph node metastasis evident during staging. CD = Clavien Dindo Classification. Staging according to UICC 8th edition. *p*-values printed in bold are considered statistically significant.

**Table 2 cancers-13-01303-t002:** Univariate and multivariate regression analysis for OS.

Univariate	HR	CI95%	*p*-Value	Multivariate	HR	CI95%	*p*-Value
Age	1.02	1.01–1.03	**<0.0001**		1.02	1.01–1.03	**<0.0001**
Siewert type II	1.00				1.00		
Siewert type III	1.20	1.03–1.39	**0.02**		1.02	0.94–1.10	0.70
Gender (Ref: female)	1.00	0.83–1.20	0.99		1.11	0.91–1.34	0.30
Neoadjuvant CTx	0.86	0.74–1.01	**0.07**		0.91	0.77–1.08	0.29
Lauren not classified	1.00		**<0.0001**		1.00		**0.01**
Lauren intestinal	1.19	0.94–1.51	0.15		1.32	1.03–1.71	**0.03**
Lauren diffuse	1.68	1.27–2.23	**<0.0001**		1.64	1.23–2.20	**0.00**
Lauren mixed	1.55	1.14–2.11	**0.01**		1.40	1.02–1.93	**0.04**
D2-dissection	0.83	0.7–0.98	**0.03**		0.90	0.76–1.07	0.24
UICC			**<0.0001**				**<0.0001**
UICC I	1.71	0.69–4.22	0.24		1.28	0.51–3.19	0.59
UICC II	4.36	1.79–10.92	**0.001**		3.19	1.3–7.81	**0.01**
UICC III	9.81	4.06–23.71	**<0.0001**		7.38	3.03–17.99	**<0.0001**
pR1 (Ref.: R0)	2.59	2.09–3.20	**<0.0001**		1.51	1.21–1.89	**<0.0001**
Grading (G1/2 vs G3/4)	1.39	1.18–1.64	**<0.0001**		1.02	0.84–1.23	0.87
Comorbidity present	1.12	0.95–1.31	0.18		1.05	0.89–1.24	0.58
Complication present	1.29	1.09–1.53	**0.004**		1.24	1.05–1.48	**0.01**

Legend: HR = Hazard Ratio, CI95% lower: 95% Confidence Interval lower boundary, CI95% upper: 95% Confidence Interval upper boundary, *p*-values printed in bold are considered statistically significant.

**Table 3 cancers-13-01303-t003:** Baseline characteristics for patients with differentiated/undifferentiated histotypes before propensity score matching (PSM).

Variable	Differentiated (*n* = 573)		Undifferentiated (*n* = 580)		*p*-Value
	*n*	%	*n*	%	
Gender					<0.0001
Female	102	17.80	154	26.55	
Male	471	82.20	426	73.45	
Age					0.256
<70	377	65.79	393	67.76	0.492
>70	196	34.21	187	32.24	
Comorbidities					0.009
None	424	74.00	393	67.76	
Single	68	11.87	106	18.28	
Multiple	81	14.14	81	13.97	
Localization					<0.0001
Siewert II	364	63.53	252	43.45	
Siewert III	209	36.47	328	56.55	
Neoadjuvant Treatment					0.145
Chemotherapy	263	45.90	234	40.34	
Chemoradiation	12	2.09	11	1.90	
None	298	52.01	335	57.76	
Type of Surgery					<0.0001
Esophagectomy	116	20.24	79	13.62	
Transhiat. ext. Gastrectomy	379	66.14	426	73.45	
Total gastrectomy	34	5.93	64	11.03	
Merendino	44	7.68	11	1.90	
Surgical extension					0.003
None	171	29.84	134	23.10	
Luminal/transhiatal	245	42.76	225	38.79	
Splenectomy	36	6.28	46	7.93	
Colon	5	0.87	3	0.52	
Pancreas	10	1.75	15	2.59	
Others	106	18.50	157	27.07	
Dissected LN (Median)	28 (3–118)		30 (3–105)		
<= 25	208	36.30	181	31.21	0.071
>25	365	63.70	399	68.79	
Complications					0.500
None	425	74.17	413	71.21	
CD I/II	77	13.44	90	15.52	
CD III-V	71	12.39	77	13.28	
pT					<0.0001
pT1	169	29.49	62	10.69	
pT2	65	11.34	79	13.62	
pT3	241	42.06	264	45.52	
pT4	98	17.10	175	30.17	
pN					<0.0001
pN0	284	49.56	200	34.48	
pN1	94	16.40	95	16.38	
pN2	92	16.06	98	16.90	
pN3	103	17.98	187	32.24	
UICC					<0.0001
UICC I	206	35.95	102	17.59	
UICC II	166	28.97	179	30.86	
UICC III	201	35.08	299	51.55	
Lauren subtype					<0.0001
Unclassified	188	32.81	0	0.00	
Intestinal	385	67.19	281	48.45	
Diffuse	0	0.00	172	29.66	
Mixed	0	0.00	127	21.90	
Grading					<0.0001
G1/G2	365	63.70	12	2.07	
G3/G4/Gx	208	36.30	568	97.93	
R					0.001
R0	525	91.62	496	85.52	
R1	48	8.38	84	14.48	

Legend: pT1 = Mucosa/Submucosa, pT2 = Muscularis propria, pT3 = Serosa, pT4 = Adjacent organs, pN0 = no lymph nodemetastasis detected during staging, pN1 = 1–2 locoregional lymph node metastasis evident, pN2 = 3–7 locoregional lymph node metastasis evident, pN3 > 15 locoregional lymph node metastasis evident during staging, LN = lymph node, CD = Clavien Dindo Classification, G1 = well differentiated, G2 = moderately differentiated, G3 = poorly differentiated, G4 = undifferentiated. R0 = resection margins free of tumor micro- and macroscopically, R1 = microscopic tumor residues in resection margin. Staging according to UICC 8th edition.

**Table 4 cancers-13-01303-t004:** Baseline characteristics for patients with differentiated/undifferentiated histotypes after PSM.

Match	Differentiated (*n* = 479)		Undifferentiated (*n* = 479)		*p*-Value
	*n*	%	*n*	%	
Gender					0.88
Female	102	21.29	105	21.92	
Male	377	78.71	374	78.08	
Age					0.100
<70	306	63.88	331	69.10	
>70	173	36.12	148	30.90	
Comorbidities					0.030
None	352	73.49	325	67.85	
Single	59	12.32	89	18.58	
Multiple	68	14.20	65	13.57	
Localization					0.08
Siewert II	270	56.37	242	50.52	
Siewert III	209	43.63	237	49.48	
Neoadjuvant Treatment					0.09
Chemotherapy	209	43.63	204	42.59	
Chemoradiation	9	1.88	10	2.09	
None	261	54.49	265	55.32	
Type of Surgery					<0.001
Esophagectomy	87	18.16	71	14.82	
Transhiat. ext. Gastrectomy	327	68.27	352	73.49	
Total gastrectomy	30	6.26	45	9.39	
Merendino	35	7.31	11	2.30	
Surgical extension					0.005
None	140	29.23	103	21.50	
Luminal/transhiatal	206	43.01	195	40.71	
Splenectomy	33	6.89	34	7.10	
Colon	5	1.04	3	0.63	
Pancreas	8	1.67	14	2.92	
Others	87	18.16	130	27.14	
Dissected LN (Median)	29 (3–218)		30 (3–105)		0.081
<= 25	159	33.19	156	32.57	0.890
>25	320	66.81	323	67.43	
Complications					0.190
None	356	74.32	338	70.56	
CD I/II	62	12.94	82	17.12	
CD III-V	61	12.73	59	12.32	
pT					<0.0001
pT1	140	29.23	58	12.11	
pT2	51	10.65	68	14.20	
pT3	200	41.75	228	47.60	
pT4	88	18.37	125	26.10	
pN					<0.0001
pN0	233	48.64	176	36.74	
pN1	75	15.66	78	16.28	
pN2	77	16.08	78	16.28	
pN3	94	19.62	147	30.69	
UICC					<0.0001
UICC I	167	34.86	93	19.42	
UICC II	136	28.39	155	32.36	
UICC III	176	36.74	231	48.23	
Lauren subtype					<0.0001
Unclassified	160	33.40	0	0.00	
Intestinal	319	66.60	251	52.40	
Diffuse	0	0.00	126	26.30	
Mixed	0	0.00	102	21.29	
Grading					<0.0001
G1/G2	303	63.26	10	2.09	
G3/G4/Gx	176	36.74	469	97.91	
R					0.67
R0	431	89.98	426	88.94	
R1	48	10.02	53	11.06	

Legend: pT1 = Mucosa/Submucosa. pT2 = Muscularis propria. pT3 = Serosa. pT4 = Adjacent organs. pN0 = no lymph nodemetastasis detected during staging pN1 = 1–2 locoregional lymph node metastasis evident, pN2 = 3–7 locoregional lymph node metastasis evident, pN3 > 15 locoregional lymph node metastasis evident during staging. CD = Clavien Dindo Classification. Staging according to UICC 8th edition.

## Data Availability

The data presented in this study are available on request from the corresponding author. The data are not publicly available due to the European General Data Protection Regulation (GDPR).

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
