# Peer review of "Novel Histologic Categorization Based on Lauren Histotypes Conveys Prognostic Information for Gastroesophageal Junction Cancers—Analysis from a Large Single Center Cohort in Germany"

_cancers, 2021, doi:10.3390/cancers13061303_

Round 1
Reviewer 1 Report
The study evaluates the prognosis impact of the Lauren classification and proposes a simplified classification for GE-junction cancers.
- The study proposed to simplify the Lauren classification to two subtypes based on patients’ survival in different Lauren subtypes. However, survival is probably only one criterium for classification of subtypes. What about patients’ response to therapies? That is, the simplified classification with two subtypes (differentiated and undifferentiated) is probably good in terms of prognosis, however, it may not be good to distinguish patients who may respond to different therapies differently and need different best therapies.
- The simplified classification works well with the patient population in the study but may not be good for more diverse patients in other parts of the world.
- Please replace “multivariate analysis” with “multiple variable analysis” in the paper since “multivariate analysis” means differently in statistics – analysis having multiple outcomes in one model instead of having multiple covariates in one model of multiple variable analysis.
- The paper does not include rationale and evaluation why a propensity score matching (PSM) is needed for their study compared to other matching methods (e.g. exact matching, coarsing matching), and option of matching vs. regression adjustment, stratification, weighting.
- After PSM and before the analysis, several evaluations are need to ensure the PSM is adequate, such as checking the covariate balance between groups before and after PSM, evaluating the improvement in covariate balance compared to the original data, assessing the overlap in covariates between groups ……
- Please replace the coma with dot in numbers.
Reviewer 2 Report
Very interesting analysis. Prognostic stratification based on histologic tumor categorization is an important issue for future personalization of treatment strategy in GEJ cancer. The manuscript is a valuable contribution on the topic an I recommend publication after minor revision.
In the revision I would recommend to include the 21 AEG1 cases to the analysis.
Three reasons:
- In a collective of 1174 patients a 21/616 /537 AEG1/2/3 ratio suggest the concusion of severe AEG classification error.
- TNM 8 grossly classifies AEG 1 and 2 as esophageal cancers. A division of AEG 1 and AEG 2 is not reasonable.
- AEG 1 and 2 are both treated by the same multimodal protocols and both increasingly treated by Ivor-Lewis Esophagectomy instead of extended gastrectomy.
